# Effectiveness of BNT162b2 after extending the primary series dosing interval in children and adolescents aged 5–17

Francisco Tsz Tsun Lai [1,2,7], Min Fan [1,7], Caige Huang[1], Celine Sze Ling Chui[2,3,4], Eric Yuk Fai Wan [1,2,5], Xue Li [1,2,6], Carlos King Ho Wong [1,2,5], Ching-Lung Cheung [1], Ian Chi Kei Wong [1,8] ✉ & Esther Wai Yin Chan [1,2,8] ✉

Extended intervals between the first and second doses of mRNA Covid-19 vaccines may reduce the risk of myocarditis in children and adolescents. However, vaccine effectiveness after this extension remains unclear. To examine this potential variable effectiveness, we conducted a population-based nested case-control study of children and adolescents aged 5–17 years who had received two doses of BNT162b2 in Hong Kong. From January 1 to August 15, 2022, 5396 Covid-19 cases and 202 Covid-19 related hospitalizations were identified and matched with 21,577 and 808 controls, respectively. For vaccine recipients with extended intervals [≥28 days, adjusted odds ratio 0.718, 95% Confidence Interval: 0.619, 0.833] there was a 29.2%-reduced risk of Covid-19 infection compared to those with regular intervals (21–27 days). If the threshold was set at eight weeks, the risk reduction was estimated at 43.5% (aOR 0.565, 95% CI: 0.456, 0.700). In conclusion, longer dosing intervals for children and adolescents should be considered.

Myocarditis has been identified as a rare side effect of messenger RNA (mRNA) Covid-19 vaccines[1,2], with individuals reported to be at an approximately threefold increased risk after BNT162b2 vaccination. Consistent, clear evidence supporting such an association was found using both a cohort[3] and a case–control design[4] in the Hong Kong population. The increased risk was mainly driven by the second dose, with a higher risk observed in young and male individuals[4], consistent with studies in other populations[5,6].

The BNT162b2 vaccination program for adolescents in Hong Kong was launched in June 2021. The age threshold was extended to ≥16 years on 15 April 2021, ≥12 on 11 June 2021, ≥5 on 21 January 2022, and subsequently, ≥6 months old on 4 August 2022[7–10]. Similar to other jurisdictions, Hong Kong took early action in extending the

recommended dosing interval between the first two doses[11], i.e., primary series, of BNT162b2 from 21 days to 56 days for adolescents under the age of 18 to reduce the risk of myocarditis on 17 June 2022[12–15]. We evaluated the immediate effect of this policy and found a significant decline in myocarditis cases[16]. While this rare BNT162b2 side effect may have been effectively reduced, there is, however, no study examining how the effectiveness of the vaccine might differ from a regular dosing interval. In the face of rapid surges of SARS-CoV-2 infection cases, it is difficult to evaluate the tradeoff between timely immunization and reduction in risk of side effects.

An early immunogenicity study in the United Kingdom reported a stronger neutralizing antibody response and enriched CD4+ T cells expressing interleukin-2 (IL-2) associated with an extended dosing

[1]Centre for Safe Medication Practice and Research, Department of Pharmacology and Pharmacy, Li Ka Shing Faculty of Medicine, The University of Hong Kong, Pokfulam, Hong Kong. [2]Laboratory of Data Discovery for Health (D24H), Hong Kong Science Park, Shatin, Hong Kong. [3]School of Nursing, Li Ka Shing Faculty of Medicine, University of Hong Kong, Pokfulam, Hong Kong. [4]School of Public Health, Li Ka Shing Faculty of Medicine, University of Hong Kong, Pokfulam, Hong Kong. [5]Department of Family Medicine and Primary Care, School of Clinical Medicine, Li Ka Shing Faculty of Medicine, University of Hong Kong, Pokfulam, Hong Kong. [6]Department of Medicine, School of Clinical Medicine, Li Ka Shing Faculty of Medicine, University of Hong Kong, Pokfulam, Hong Kong. [7]These authors contributed equally: Francisco Tsz Tsun Lai, Min Fan. [8]These authors jointly supervised this work: Ian Chi Kei Wong, Esther Wai Yin Chan. ✉e-mail: wongick@hku.hk; ewchan@hku.hk

interval[17]. In a Canadian test-negative case–control study of adults aged 18 or above, higher vaccine effectiveness has also been identified after an extended dosing interval[18]. In this study, accordingly, we hypothesize greater effectiveness of BNT162b2 associated with extended dosing intervals between the first two doses, compared with regular intervals (21–27 days). We tested this hypothesis in children and adolescents in Hong Kong with a nested case–control study using a territory-wide public healthcare database.

## Results

Figure 1 illustrates the selection of cases and controls from the underlying cohort. A total of 5396 Covid-19 infection cases and 202 Covid-19 related hospitalizations within 28 days of infection were identified, matched with 21,577 and 808 controls from 1 January to 15 August 2022. 3.0% (653/21577) of controls in the Covid-19 matched sets, and 0.1% (1/807) controls in Covid-19 related hospitalizations matched sets subsequently developed into cases. The baseline characteristics of cases and controls are shown in Table 1, with clinical and other characteristics largely similar between cases and controls. Detailed demographic information stratified by both case and exposure status is shown in Supplementary Table 1. Supplementary Table 2 tabulates the number of cases and controls by age group.

According to the multivariable model of Covid-19 infection as an outcome (Table 2), children or adolescents with an extended dosing interval (28 days or above) had a 29.2%-risk deduction [adjusted odds ratio (aOR) 0.718, 95% confident interval (CI): 0.619, 0.833], compared with regular intervals (21–27 days). Consistent results were observed between sexes with an estimated Covid-19 risk reduction of 35.4% for females (aOR 0.646, 95% CI: 0.519, 0.803) and 25.8% for males (aOR 0.742, 95% CI: 0.610, 0.902). The estimation for risk of Covid-19-related hospitalization was not statistically significant (aOR 0.743, 95% CI: 0.338, 1.636). A subgroup analysis for adolescents between 12-17 years old estimated a similar odds ratio with the primary analysis, for both Covid-19 risk (aOR 0.724, 95% CI: 0.624, 0.840) and Covid −19-related hospitalization (aOR 0.835, 95% CI: 0.360,1.939).

Sensitivity analyses (Table 2) largely support the robustness of the main results. When 56 days were used as the cutoff for the definition of an extended dosing interval, there was a larger risk reduction of infection, i.e., 43.5% (aOR 0.565, 95% CI: 0.456, 0.700). The estimation remained similar after excluding children and adolescents with extreme dosing intervals (aOR 0.762, 95% CI: 0.653, 0.889); and adding rapid antigen test (RAT) positive cases to the definition of Covid-19 (aOR 0.829, 95% CI: 0.745, 0.923). Greater protection for Covid-19 infection was observed with a further prolongation of the dosing interval [aOR:0.848, 95% CI: 0.715, 1.006 for 28–55 days, aOR: 0.652, 95% CI: 0.485, 0.877 for 56–83 days, and aOR:0.540, 95% CI: 0.435, 0.672 for 84 days or above]. A similar protective effect was evident when examining only patients with extended dosing intervals who received their second dose 21 days or more before the policy change (aOR: 0.690, 95% CI: 0.593, 0.802 for covid-19 infection and aOR: 0.716, 95% CI:0.313, 1.638 for covid-19-related hospitalization).

## Discussion

These population-based data from Hong Kong support our hypothesis that BNT162b2 has greater effectiveness against SARS-CoV-2 infection in children and adolescents if the primary doses are administered over an extended interval of four weeks or more, as opposed to only three weeks or shorter. Further increase in effectiveness was observed in thresholds set at eight or 12 weeks, with larger reductions in the odds of infection. No significant difference in Covid-19 hospitalization outcomes was detected.

To the best of our knowledge, this is the first population-based study investigating the real-world effectiveness of BNT162b2 with the dosing interval of the primary series extended for children and adolescents. Our findings are consistent with the aforementioned immunogenicity study of 589 individuals which shows a stronger neutralizing antibody response and an enrichment of CD4+ T cells with an extended dosing interval (17), although in the current study, differences in Covid-19 hospitalization as an outcome were not significant. This is in line with another study of 93 healthcare professionals showing a stronger humoral response with the dosing

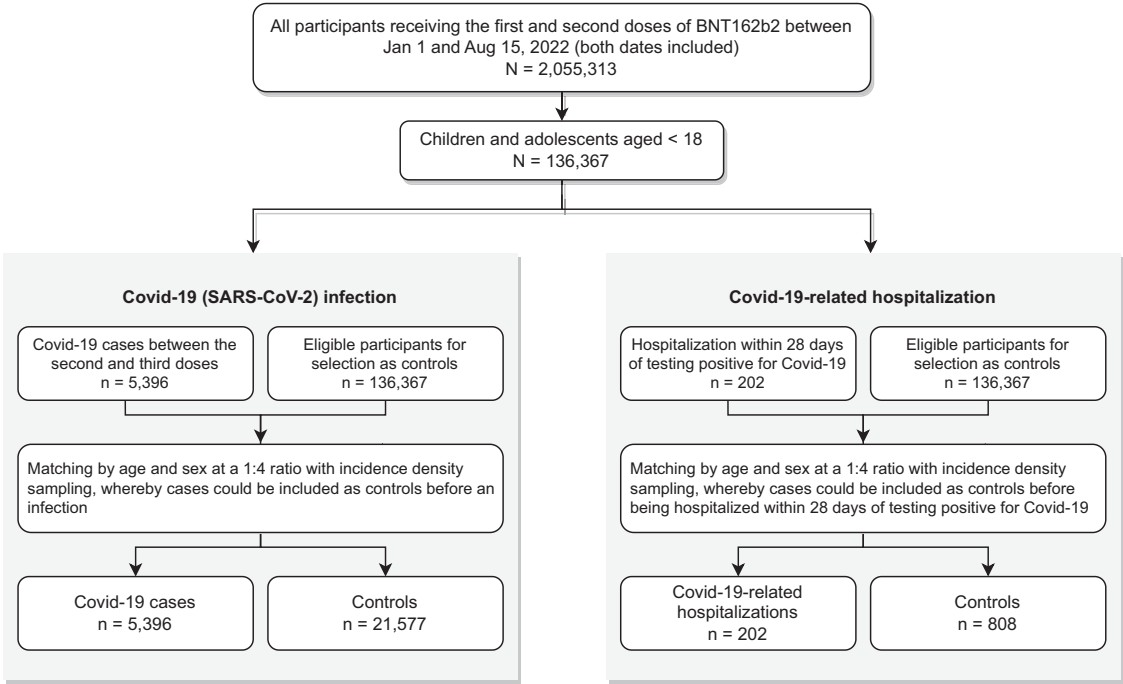

**Fig. 1 | Flowchart of cases and controls selection.** Flowchart illustrating the selection procedures of cases and controls for the assessment of vaccine effectiveness after extending the dosing intervals between the first and second doses of

BNT162b2. SARS-CoV-2 severe acute respiratory syndrome coronavirus 2; Covid-19 coronavirus disease 2019.

**Table 1 | Characteristics of cases and controls in two case-control studies for Covid-19 infection and Covid-19 hospitalization**

| | Covid-19 infection | | | Covid-19 hospitalization | | |
|---|---|---|---|---|---|---|
| | Control N = 21577 | Case N = 5396 | SMD | Control N = 808 | Case N = 202 | SMD |
| Extended dosing interval (%) | 5518 (25.6) 191.00 | 1248 (23.1) 189.00 | 0.057 | 261 (32.3) 164.00 | 63 (31.2) 167.00 | 0.024 |
| Onset time [days, median (25–75 percentile)] | [158.00, 213.00] | [164.00, 211.00] | 0.033 | [98.00, 199.00] | [107.25, 194.75] | 0.001 |
| Age [mean (SD)] | 15.07 (1.77) | 15.06 (1.78) | 0.001 | 14.69 (2.25) | 14.69 (2.25) | <0.001 |
| Sex (male,%) | 11584 (53.7) | 2897 (53.7) | <0.001 | 420 (52.0) | 105 (52.0) | <0.001 |
| Pre-existing comorbidities (%) | | | | | | |
| Asthma | 258 (1.2) | 96 (1.8) | 0.048 | 11 (1.4) | 5 (2.5) | 0.081 |
| Diabetes | 35 (0.2) | 8 (0.1) | 0.004 | 1 (0.1) | 0 (0.0) | 0.050 |
| Epilepsy | 160 (0.7) | 55 (1.0) | 0.030 | 13 (1.6) | 10 (5.0) | 0.188 |
| The use of immunosuppressants within 90 days before index date (%) | 19 (0.1) | 3 (0.1) | 0.012 | 0 (0.0) | 2 (1.0) | 0.141 |

Standard mean difference (SMD), standard deviation (SD), Covid-19 (Coronavirus disease 2019).

interval extended, but T cell response was shown to be comparable with a regular dosing interval[19]. Indeed, while humoral response correlates consistently with reduced SARS-CoV-2 infections[20], cellular response correlates more specifically with a reduced risk of severe Covid-19 disease[21]. Another possible reason for this observation could simply be the modest sample size. Of note, a few other studies on the real-world effectiveness between regular and extended dosing intervals showed similar results to our study[18,22,23], however, children or adolescents were not represented in these studies.

Our study revisited the question of whether to recommend delaying the second dose[24], however, this has generated a different rationale to the initial objective of an earlier first-dose vaccination for the greatest number of people. For the adult population, assuming a sufficient supply of vaccines and effective implementation of vaccine roll-outs, there is no compelling reason to delay the second priming dose of BNT162b2 since the incidence of myocarditis or other side effects are rare[4,25] and timely completion of immunization is much preferred. In children and adolescents, although rare, the risk of myocarditis after vaccination is evident[26]. Measures should be taken to reduce the risk of this iatrogenic condition. Our previous work has demonstrated that delaying a second dose is safe with no increased risk of adverse events of special interest[27]. This study shows that, in addition to the potential reduction of the risk of myocarditis, vaccine effectiveness of BNT162b2 is also strengthened with an extended dosing interval in children and adolescents. Therefore, our findings firmly support the extension of dosing intervals for children and adolescents in future mass mRNA vaccinations.

This study's strengths include the population-based sample which should confer a high representativeness of the findings, as well as the territory-wide unified recording system used by our data providers. There are, however, several limitations that warrant caution. First, this was an observational study without randomization to eliminate potential selection bias. Unmeasured confounding or indication bias might affect the results. Second, similar with other observational studies, the list of covariates for multivariable adjustment was limited by sample size and data availability. Future studies should consider a broader inclusion of different covariates such as lifestyle factors. Third, the ethnicity of the sample was predominantly Chinese and the generalizability of the findings should be tested in other populations. Fourth, as this is a case–control study, only relative but not absolute risks were estimated. However, as children with the extended dosing interval typically received the vaccine at a later date, a cohort study comparing groups exposed to different periods of the pandemic would have been challenging in terms of methodology. Fifth, pediatric formulations were used specifically for children aged younger than 12 years, which indicates a dosage difference in children aged 5-11 and 12-17. However, our modest sample size does not allow a stratified

analysis by age group. Further analysis with larger samples is required. It is also possible that incidental positives were included in the Covid-19 hospitalization cases. However, our current definition of Covid-19-related hospitalization is also the most widely used approach and in view of the young age of our target population, this should not influence our estimation. Fifth, there may be potential differences in various unmeasured characteristics between families who chose the regular dosing interval and those who chose the extended dosing interval by self-selection. Last, we do not have data on the specific variants down to the individual level, but during our sampling period, the Omicron variant was the dominant strain in Hong Kong therefore our findings can be reasonably generalized to this particular variant.

In conclusion, in this population-based nested case–control study, we identified greater effectiveness of BNT162b2 with an extended dosing interval between the first two priming doses, compared with a regular interval. To reduce the risk of myocarditis, an extended dosing interval for mRNA vaccines should be considered for children and adolescents.

## Methods

We established a Covid-19 database in Hong Kong by matching electronic health records from the Hospital Authority (HA), vaccination records from the Department of Health (DH), and Covid-19-confirmed case records from the Centre of Health Protection (CHP) based on deidentified unique pseudo-identifiers. The database has been widely used to evaluate Covid-19 vaccine safety, effectiveness, and long Covid outcomes[28–31]. The HA provides publicly funded health services to over 7.4 million Hong Kong residents, with 43 public hospitals, 49 specialist outpatient clinics, and primary care clinics[32,33]. They maintain electronic clinical records of diagnosis, prescription, laboratory results, emergency department attendance, and hospitalization details. The DH operates all Covid-19 mass vaccination programs in Hong Kong. The CHP maintains a database of confirmed Covid-19 cases, including both mandatory and voluntary reporting of positive polymerase chain reaction (PCR) and RAT results. RAT kits were widely available in Hong Kong at a generally affordable price. The Government also distributed free RAT kits occasionally and families often kept test kits at home in case of suspected infection of household members. Individuals testing positive using RAT were required to report to the CHP and at the earlier stages of the outbreak when daily PCR test capacity was sufficient, a mandatory PCR test would follow to confirm the case. From 7 March 2022 onwards, RAT-positive individuals reporting to the Centre are considered confirmed cases with only a proportion randomly selected for subsequent confirmatory PCR tests.

From December 2020, the government began surveillance of SARS-CoV-2 variants by whole genome sequencing from sampled cases. The predominant circulating variant during the study period was Omicron[34].

**Table 2 | The risk of Covid-19 infection and hospitalization in children and adolescents with extended dosing intervals compared with regular dosing intervals**

| | Crude OR (95% CI) | Adjusted OR (95% CI)$ |
|---|---|---|
| *Covid-19 infection* | | |
| Primary analysis | 0.717 (0.618, 0.832) | 0.718 (0.619, 0.833) |
| Subgroup analysis by sex | | |
| Male | 0.742 (0.610, 0.903) | 0.742 (0.610, 0.902) |
| Female | 0.643 (0.517, 0.799) | 0.646 (0.519, 0.803) |
| Subgroup analysis for adolescents between ages 12–17 | 0.723 (0.623, 0.838) | 0.724 (0.624, 0.840) |
| Sensitivity analysis | | |
| Using 56 days as the exposure cutoff | 0.564 (0.455, 0.699) | 0.565 (0.456, 0.700) |
| Removing participants with extreme intervals | 0.760 (0.652, 0.887) | 0.762 (0.653, 0.889) |
| Adding RAT-positive testing | 0.829 (0.745, 0.922) | 0.829 (0.745, 0.923) |
| Adding additional dosing interval exposure group | | |
| 28–55 days | 0.846 (0.713–1.004) | 0.848 (0.715-1.006) |
| 56–83 days | 0.654 (0.487–0.879) | 0.652 (0.485–0.877) |
| 84 days or above | 0.540 (0.435–0.672) | 0.540 (0.435–0.672) |
| Adding the timing of policy change as a potential confounder | | |
| Regular dosing interval: Received second dose 21 days or more before policy change | Ref. | Ref. |
| Regular dosing interval: Received second dose within 21 days before or after policy change† | | |
| Extended dosing interval: Received second dose 21 days or more before policy change | 0.689 (0.593–0.802) | 0.690 (0.593–0.802) |
| Extended dosing interval: Received second dose within 21 days before or after policy change | 0.399 (0.282–0.564) | 0.397 (0.281–0.561) |
| *Covid-19 hospitalization* | | |
| Primary analysis | 0.691 (0.315, 1.516) | 0.743 (0.338, 1.636) |
| Subgroup analysis by sex | | |
| Male | 0.883 (0.282, 2.769) | 0.972 (0.305, 3.097) |
| Female | 0.638 (0.212, 1.924) | 0.687 (0.224, 2.104) |
| Subgroup analysis for adolescents between ages 12–17 | 0.845 (0.370, 1.929) | 0.835 (0.360, 1.939) |
| Sensitivity analysis | | |
| Using 56 days as the cutoff | 0.594 (0.210, 1.676) | 0.637 (0.223, 1.820) |
| Removing adolescents with extreme intervals | 0.729 (0.326, 1.630) | 0.790 (0.353, 1.768) |
| Adding RAT-positive testing | 0.732 (0.316, 1.695) | 0.779 (0.336, 1.805) |
| Adding additional dosing interval exposure group | | |
| 28–55 days | 0.817 (0.323–2.062) | 0.656 (0.224–1.925) |
| 56–83 days | 1.107 (0.297–4.121) | 1.632 (0.417–6.385) |
| 84 days or above | 0.537 (0.184–1.572) | 0.742 (0.240–2.290) |
| Adding the timing of policy change as a potential confounder | | |
| Regular dosing interval: received second dose 21 days or more before policy change | Ref. | Ref. |
| Regular dosing interval: received second dose within 21 days before or after policy change† | | |
| Extended dosing interval: received second dose 21 days or more before policy change | 0.706 (0.318–1.568) | 0.716 (0.313–1.638) |
| Extended dosing interval: received second dose within 21 days before or after policy change | 0.791 (0.220–2.849) | 0.781 (0.211–2.894) |

$The estimation is after adjusting pre-existing asthma, diabetes, epilepsy, onset time, and immunosuppressant use within 90 days.
†No participants received the second dose within a regular dosing interval but also within 21 days before the policy change or after it.
Odds ratio (OR), confident interval (CI), and Covid-19 (Coronavirus disease 2019).

## Study design and case−control selection

This is a nested case−control study estimating the BNT162b2 effectiveness between regular (21–27 days) and extended (28 days or above) dosing intervals of the priming doses. The underlying cohort includes all children and adolescents aged three to 17 years who received both the first and second doses of BNT162b2 without any prior SARS-COV-2 infections. The period for case and control selection spanned from 1 January to 15 August 2022.

The evaluation of vaccine effectiveness included (1) Covid-19 infection, defined as the first recorded SARS-CoV-2 positive PCR; and (2) the first Covid-19-related hospitalization within 28 days after SARS-CoV-2 infection. All observation periods were censored upon the third dose of vaccination. PCR is widely considered the gold standard for the clinical definition of a SARS-CoV-2 infection[35]. RAT

results were not considered in the primary analysis since they are self-reported and deemed far less reliable than a PCR test administered by trained personnel from the CHP. The index date for cases was the infection date (date of positive test result) or hospital admission date, respectively. Within the underlying cohort, all participants, before categorization as a case, death, or receiving the third dose, were eligible to be sampled as controls. Using incidence density sampling, up to four controls were randomly selected and matched with each case by age and sex. By using incidence density sampling, all participants in the same matched set were exposed to the same degree of Covid-19 risk and were under the same policy guidance. The index date of controls was defined as their matched cases' index date. Controls could be matched to multiple cases or be selected as a case in later periods.

## Statistical analysis

We estimated the odds ratio (OR) with 95% confidence interval by conditional logistic regression. The adjusted odds ratio (aOR) was evaluated by adjusting onset time (day from the second dose to index date), pre-existing comorbidities dating back as early as 2016, and the use of immunosuppressant within 90 days before the index date. The comorbidities included asthma, diabetes, and epilepsy, which were chosen based on local disease epidemiology in consultation with pediatricians. Subgroup analysis by sex and among those aged 12–17 were conducted. Three sensitivity analyses were conducted by (1) changing the cutoff of dosing interval from 28 to 56 days; (2) removing the vaccine recipients with extreme higher dosing intervals (80th-percentile or above); (3) including those with RAT-positive cases; (4) The exposure group were separated into 28–55 days (4–7 weeks), 56–83 days (8–11 weeks), and 84 days or above (12 weeks or above) to identify any dose-response relationship; and (5) The exposure statuses were further categorized by dosing interval (extended versus regular) and whether the second dose was received 21 days or more before the policy change to identify any potential confounding effects from the policy change, independent of the effect from the modified regimen.

A detailed sample size calculation is attached in Supplementary Figure. A $P$ value of 0.05 or lower was considered indicative of statistical significance. All analyzes were performed in R version 4.1.0 (R Foundation for Statistical Computing, Vienna, Austria). Results were conducted independently by two researchers (M.F. and C.H.).

## Ethics approval

This study was approved by the Central Institutional Review Board of the Hospital Authority of Hong Kong (CIRB-2021-005-4) and the Department of Health Ethics Committee (LM171/2021). As our data were all anonymized without any personal identification information, no informed consent was required for the study.

## Reporting summary

Further information on research design is available in the Nature Portfolio Reporting Summary linked to this article.

## Data availability

Data used for this study will not be available to others as the data custodians have not given permission due to concerns over patient privacy protection. Requests for data access could be submitted to the Central Panel on Administrative Assessment of External Data Requests of the Hospital Authority (hacpaaedr@ha.org.hk). As the data provided will be customized for the specific purpose of each project, the time duration required to process such requests may vary. Upon data request approval, no sharing of such data with third parties is allowed.

## Code availability

The code used for this study is available on the GitHub and Zenodo.

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

## Acknowledgements

This was a regulatory pharmacovigilance study initiated by the Department of Health and funded via the Health Bureau of the Government of the Hong Kong Special Administrative Region (COVID1903011). F.T.T.L. and I.C.K.W. are partially supported by the Laboratory of Data Discovery for Health (D24H) funded by AIR@InnoHK administered by the Innovation and Technology Commission. We gratefully acknowledge the Hospital Authority and the Department of Health for the provision of data and Lisa Y Lam for proofreading the manuscript.

## Author contributions

F.T.T.L., M.F., and E.W.Y.C. had the original idea for the study, contributed to the development of the study, extracted data from the source database, constructed the study design and the statistical model, reviewed the literature, and act as guarantors for the study. M.F. and C.H. undertook the statistical analysis. F.T.T.L. and M.F. wrote the first draft of the manuscript. F.T.T.L., I.C.K.W., and E.W.Y.C. are the principal investigators and provided oversight for all aspects of this project. C.S.L.C., E.Y.F.W., X.L., C.K.H.W., and C.-L.C. provided critical input to the analyses, design, and discussion. All authors contributed to the interpretation of the analysis, critically reviewed and revised the manuscript, and approved the final manuscript as submitted. F.T.T.L., M.F., C.H., I.C.K.W., and E.W.Y.C. have accessed and verified the data used in the study. All authors had full access to all the data in the study and had final responsibility for the decision to submit for publication.

## Competing interests

F.T.T.L. has been supported by the RGC Postdoctoral Fellowship under the Hong Kong Research Grants Council and has received research grants from the Health Bureau of the Government of the Hong Kong Special Administrative Region, outside the submitted work. CSLC has received grants from the Health Bureau of the Hong Kong Government, Hong Kong Research Grant Council, Hong Kong Innovation and Technology Commission, Pfizer, IQVIA, MSD, and Amgen; and personal fees from PrimeVigilance; and grants from Research Grants Council (RGC/ ECS, Hong Kong), outside the submitted work. E.Y.F.W. has received research grants from the Health Bureau of the Government of the Hong Kong Special Administrative Region, and the Hong Kong Research Grants Council, outside the submitted work. X.L. has received research grants from the Health Bureau of the Government of the Hong Kong Special Administrative Region; research and educational grants from Janssen and Pfizer; internal funding from the University of Hong Kong; consultancy fees from Merck Sharp & Dohme; and grants from Research Grants Council (RGC/ECS, Hong Kong), unrelated to this work. C.K.H.W. reports receipt of research funding from the EuroQoL Group Research Foundation, the Hong Kong Research Grants Council, and the Hong Kong Health and Medical Research Fund; outside of the submitted work. C.L.C. received research grants and honorarium from Amgen, research grant support from HMRF, and honorarium from Abbott. I.C.K.W. reports research funding outside the submitted work from Amgen, Bristol-Myers Squibb, Pfizer, Janssen, Bayer, GSK, Novartis, the Hong Kong Research Grants Council, the Health Bureau of the Government of the Hong Kong Special Administrative Region, National Institute for Health Research in England, European Commission, and the National Health and Medical Research Council in Australia; has received speaker fees from Janssen and Medice in the previous 3 years; and is an independent non-executive director of Jacobson Medical in Hong Kong. E.W.Y.C. reports honorarium from Hospital Authority; and grants from the Research Grants Council (RGC, Hong Kong), Research Fund Secretariat of the Health Bureau, National Natural Science Fund of China, Wellcome Trust, Bayer, Bristol-Myers Squibb, Pfizer, Janssen, Amgen, Takeda, and Narcotics Division of the Security Bureau of the Hong Kong Special Administrative Region, outside the submitted work. All other authors declare no competing interests.
