## [Peer Review File · Nature Communications]

Effectiveness of BNT162b2 after extending the primary-series dosing interval in children and adolescents aged 5-17REVIEWER COMMENTS

Reviewer #1 (Remarks to the Author):

The authors used a population-based nested case control study to examine the impact of extended intervals between priming doses on BNT vaccine effectiveness in children and adolescents. The rich data source is a strength of the study, which adds to the pediatric COVID-19 VE literature; the question is an important one, given the potential decrease in myocarditis risk using an extended interval.

While the data support the conclusions, some methodologic details are lacking or need to be clarified. Please see my general and specific comments below.

GENERAL COMMENTS

- What was the availability of home-based RAT in the population over the study period? How comprehensive is the CHR testing?
- Did the authorized vaccine dosages differ across the study population? If yes, are age stratified analyses appropriate?
- Confirm the circulating variant(s) across the study period. This is currently stated in the discussion, but should be included in the methods.
- Was the underlying cause of the COVID-19 hospitalizations confirmed to be the infection, or could they have been incidental positives?
- Were prior SARS-CoV-2 infections examined?

SPECIFIC COMMENTS

Lines 145-147: (1) The adolescent program launched in June, but extended in April? (2) See comment below regarding the <5 year olds.

Line 154: Per line 147, the 3- and 4-year-olds would not have been eligible for vaccination until August 4. Please clarify how those ages were included in the study population.

Line 160: Infection date = date of positive test result?

Line 169: Use of immunosuppressants in the 90 days before or after index date?

Line 170: Were the comorbidities identified based on diagnosis codes in the HA database? Was there a time restriction to these diagnoses?

Figure 1: Is the 202 hospitalizations a subset of the 5396 cases?

Table 1: (1) Needs footnotes/definitions, (2) which sex is displayed?

Reviewer #2 (Remarks to the Author):

Thank you for asking me to review this manuscript. The aim of this study is to estimate the protection of the BNT162b2 vaccine against SARS-CoV-2 infection and COVID-19 hospitalization conferred by extended dosing intervals between the first two doses compared with regular intervals (21-27 days). The possibility to extend the vaccine schedule without negatively affecting the vaccine effectiveness is a relevant public health topic. Moreover, this is the first study that analyses the real-world impact of a time extension between the first two doses in children and adolescents.

I believe that this paper makes a valuable contribution to the literature, however, there are a number of suggestions that will improve the clarity, readability and robustness of the results.

In Figure 1 and line 161 there is a reference to the third dose. I suggest to clarify in the introduction the development of the vaccine rollout for the children specifying when it started and when was authorized the first booster for the children. Moreover, in the methods section, it is essential to clarify how the children are treated once have received the first booster dose. Are they excluded from the analysis or the administration date is used as the index date?

In the matching procedure only two confounders are taken into account: sex and age. I would suggest matching also by the week of the administration of the first and second dose since this could be an important confounder in your analysis and could affect the final results.

An important point is the use of the hospitalization date as the index date. I would suggest using the infection date as the index date for the hospitalized cases. In the estimation of the protection against COVID-19 hospitalization, it is important to include all the infections resulting in hospital admission within 28 days in the study period. Using the hospitalization date, you could risk defining as hospitalized case individuals infected before the 1st January and on the other hand, you would risk excluding from the case definition individuals who developed the infection before the end of the study (15 August).

Please, clarify if individuals previously infected are excluded from the underlying cohort.

Please, detail the flowchart including the number of individuals that move from the control to the case group.

I would suggest changing the caption of the tables because they do not fully describe the content of the tables

I found several typos and writing mistakes I would suggest asking a native English speaker to review the manuscript

Reviewer #3 (Remarks to the Author):

This article addresses an important topical issue of what is the optimal dosing interval of COVID-19 vaccines for children and adolescent. The insights provided by this article provides guidance to policy makers in setting optimal dosing intervals

There are 2 main weaknesses in the use of observational data which exploits a potential policy change.

The first is an issue of selection. Those who chose to receive their vaccinations earlier where the 28 dose-interval was in force, are likely those who think that their children are at higher risk of contracting COVID. The results that the authors are picking up may entirely be due to the higher exposure risk of those who chose to be vaccinated earlier. The article should be clearer about whether when there was a chance of policy of moving from a 28 day to 56 dose recommended dosing interval. It would be useful to highlight what was the vaccination rates at various points of policy shifts. This would allow the reader to assess the degree of selection bias that might be driving the results.

Second, given that those who had a 28-day dose interval received their vaccinations earlier, the authors may simply be picking up waning in vaccine effectiveness, rather than the better performance of an extended dosing intervals. Here too, authors ought to match by time from second dose (i.e. i.e. difference between the index date and the date of second dose). If this is not possible, then the time from second dose ought to be reported in summary statistics for the cases and controls, for different dose intervals.

Thank you very much for considering our manuscript. We are very grateful for the insightful reviews relayed from the referees. Please find appended below, our detailed response to each of their specific comments with proposed changes and additional analyses. The revised relevant text is quoted accordingly:

Reviewer #1

- 1.1. “The authors used a population-based nested case control study to examine the impact of extended intervals between priming doses on BNT vaccine effectiveness in children and adolescents. The rich data source is a strength of the study, which adds to the pediatric COVID-19 VE literature; the question is an important one, given the potential decrease in myocarditis risk using an extended interval. While the data support the conclusions, some methodologic details are lacking or need to be clarified. Please see my general and specific comments below.”**

Author Response:

Thank you very much for your encouraging comment on the importance and key strengths of our work.

- 1.2. “What was the availability of home-based RAT in the population over the study period? How comprehensive is the CHR testing?”**

Author Response:

Thank you for this question. During the study period, RAT kits were widely available in Hong Kong at a generally affordable price. The Government also distributed free RAT kits occasionally and families often kept test kits at home in case of suspected infection of household members. Individuals testing positive using RAT were required to report to the Centre for Health Protection and at the earlier stages of the outbreak when daily PCR test capacity was sufficient, a mandatory PCR test would follow to confirm the case. From March 7, 2022, RAT-positive individuals reporting to the Centre were considered confirmed cases with a proportion randomly selected

Response to Reviewer Comments

Manuscript Title: *Effectiveness of BNT162b2 after extending the primary-series dosing interval in children and adolescents aged 3-17*
Journal: *Nature Communications*
Ref. No.: *NCOMMS-23-00794-T*

for subsequent confirmatory PCR tests. However, there were reports of falsified positive RAT results and thus cases without a PCR test for confirmation were included only in our sensitivity analysis rather than the main results. We have now added this information in the Methods section to provide a clearer picture for the readers.

“RAT kits were widely available in Hong Kong at a generally affordable price. The Government also distributed free RAT kits occasionally and families often kept test kits at home in case of suspected infection of household members. Individuals testing positive using RAT were required to report to the CHP and at the earlier stages of the outbreak when daily PCR test capacity was sufficient, a mandatory PCR test would follow to confirm the case. From March 7, 2022, RAT-positive individuals reporting to the Centre were considered confirmed cases with a proportion randomly selected for subsequent confirmatory PCR tests.”

(Lines 156-162, P. 7)

1.3. **“Did the authorized vaccine dosages differ across the study population? If yes, are age stratified analyses appropriate?”**

Author Response:

Thank you for pointing this out. There is indeed a dosage difference across age groups. Pediatric formulations were used specifically for children aged younger than 12 years. Our modest sample size does not, however, allow for a stratified analysis separately for children aged 3-11 and those aged 12 -17, with infinitely wide confidence intervals returned. We have now acknowledged this potential limitation and included it in the Limitations section.

“Fifth, pediatric formulations were used specifically for children aged younger than 12 years, which indicates a dosage difference in children aged 3-11 and 12-17. However, our modest sample size does not allow a stratified analysis by age groups. Further analysis with larger samples is required.”

(Lines 130-133, P. 6)

1.4. **“Confirm the circulating variant(s) across the study period. This is currently stated in the discussion, but should be included in the methods.”**

Author Response:

Thank you for this comment. We have now also included the circulating variant of SARS-CoV-2, i.e., Omicron, in the Methods section. We have also cited publicly

Response to Reviewer Comments

Manuscript Title: *Effectiveness of BNT162b2 after extending the primary-series dosing interval in children and adolescents aged 3-17*
Journal: *Nature Communications*
Ref. No.: *NCOMMS-23-00794-T*

available documents issued by the Centre for Health Protection to support this newly included information.

“From December 2020, the government began surveillance of SARS-CoV-2 variants by whole genome sequencing from sampled cases. The predominant circulating variant during the study period was reported to be Omicron. [1] ”

(Lines 163-165, P. 8)

1.5. **“Was the underlying cause of the COVID-19 hospitalizations confirmed to be the infection, or could they have been incidental positives?”**

Author Response:

Thanks for your question. Yes, it is possible that incidental positives were included in the COVID-19 hospitalization cases. However, considering the patients’ very young age, inclusion of incidental positives is much less likely than if the same analysis was conducted in older age groups. Moreover, our current definition of COVID-19-related hospitalization is also the most widely used approach in studying the effectiveness of COVID-19 vaccines. We have now included this issue in the Limitations section to remind readers to interpret results with caution.

“It is also possible that incidental positives were included in the Covid-19 hospitalization cases. However, our current definition of Covid-19-related hospitalization is also the most widely used approach and in view of the young age of our target population, this should not influence our estimation.”

(Lines 133-135, P. 6)

1.6. **“Were prior SARS-CoV-2 infections examined?”**

Author Response:

Thank you for this question. We removed everyone with a history of SARS-CoV-2 infection from the underlying cohort, and therefore all included infection cases were first recorded in their lifetime. We have now stated this approach more explicitly in the Methods section.

“The underlying cohort includes all children and adolescents aged three to 17 years who received both the first and second doses of BNT162b2 without any prior SARS-COV-2 infections”

(Lines 168-170, P. 8)

Response to Reviewer Comments

Manuscript Title: *Effectiveness of BNT162b2 after extending the primary-series dosing interval in children and adolescents aged 3-17*

Journal: *Nature Communications*

Ref. No.: *NCOMMS-23-00794-T*

- 1.7. “Lines 145-147: (1) The adolescent program launched in June, but extended in April? (2) See comment below regarding the <5 year olds. Line 154: Per line 147, the 3- and 4-year-olds would not have been eligible for vaccination until August 4. Please clarify how those ages were included in the study population.”**

Author Response:

Thank you for this comment. All participants aged between 3 and 17 years old were eligible for inclusion in the underlying cohort. However, no children aged 3- or 4 were eventually included in the analysis during the study period. That may be because the licensure of vaccination was only extended to this age group at a much later time in our study period: from August 4 to August 15. We have now tabulated the number of cases and controls by age groups more clearly.

Age	Covid-19 infection		Covid-19 related hospitalization	
	control	cases	control	cases
3	0	0	0	0
4	0	0	0	0
5	20	5	4	1
6	73	19	4	1
7	104	26	8	2
8	77	20	16	4
9	124	31	4	1
10	144	36	8	2
11	148	37	16	4
12	219	55	16	4
13	2032	508	92	23
14	4340	1085	148	37
15	4560	1140	172	43
16	4708	1177	132	33
17	5028	1257	188	47
Sum	21577	5396	808	202

“Appendix 2 tabulates the number of cases and controls by age group.”

(Lines 69-70, P. 4)

- 1.8. “Line 160: Infection date = date of positive test result?”**

Author Response:

Yes, the infection date was defined as the date of the positive test result.

Response to Reviewer Comments

Manuscript Title: *Effectiveness of BNT162b2 after extending the primary-series dosing interval in children and adolescents aged 3-17*
Journal: *Nature Communications*
Ref. No.: *NCOMMS-23-00794-T*

“The index date for cases was the infection date (date of positive test result) or hospital admission date, respectively.”

(Lines 177-178, P. 8)

1.9. “Line 169: Use of immunosuppressants in the 90 days before or after index date?”

Author Response:

Thank you for this question. We have now stated more clearly it was 90 days before the index date.

“The adjusted odds ratio (aOR) was evaluated by adjusting onset time (day from the second dose to index date), pre-existing comorbidities dating back as early as 2016, and the use of immunosuppressant within 90 days before the index date.”

(Lines 176-183, P. 6)

1.10. “Line 170: Were the comorbidities identified based on diagnosis codes in the HA database? Was there a time restriction to these diagnoses?”

Author Response:

Thank you for this question. The comorbidities were indeed identified using the diagnostic codes in the HA database. We have records which date back as early as year 2016. We have now stated this information more clearly in the Methods section.

“The adjusted odds ratio (aOR) was evaluated by adjusting onset time (day from the second dose to index date), pre-existing comorbidities dating back as early as 2016, and the use of immunosuppressant within 90 days before the index date.”

(Lines 186-189, P. 9)

1.11. “Figure 1: Is the 202 hospitalizations a subset of the 5396 cases?”

Author Response:

Thank you for this question. It is correct.

1.12. “Table 1: (1) Needs footnotes/definitions, (2) which sex is displayed?”

Response to Reviewer Comments

Manuscript Title: *Effectiveness of BNT162b2 after extending the primary-series dosing interval in children and adolescents aged 3-17*
Journal: *Nature Communications*
Ref. No.: *NCOMMS-23-00794-T*

Author Response:

Thank you for pointing this out. We have now edited the table to show the specific sex the numbers represent.

Reviewer #2

- 2.1. **“Thank you for asking me to review this manuscript. The aim of this study is to estimate the protection of the BNT162b2 vaccine against SARS-CoV-2 infection and COVID-19 hospitalization conferred by extended dosing intervals between the first two doses compared with regular intervals (21-27 days). The possibility to extend the vaccine schedule without negatively affecting the vaccine effectiveness is a relevant public health topic. Moreover, this is the first study that analyses the real-world impact of a time extension between the first two doses in children and adolescents. I believe that this paper makes a valuable contribution to the literature, however, there are a number of suggestions that will improve the clarity, readability and robustness of the results.”**

Author Response:

Thank you very much for the positive overall comment on our manuscript.

- 2.2. **“In Figure 1 and line 161 there is a reference to the third dose. I suggest to clarify in the introduction the development of the vaccine rollout for the children specifying when it started and when was authorized the first booster for the children.”**

Author Response:

Thank you for this comment. We have specified the dates on which the vaccination roll-out was extended to the various age groups in the Methods section.

“The BNT162b2 vaccination program for adolescents was launched in June 2021. The age threshold was extended to ≥ 16 years on April 15, 2021, ≥ 12 on June 11, 2021 ≥ 5 on January 21, 2022 and, subsequently, ≥ 6 months old on August 4, 2022. [2-5] ”

(Lines 152-154, P. 7)

Response to Reviewer Comments

Manuscript Title: *Effectiveness of BNT162b2 after extending the primary-series dosing interval in children and adolescents aged 3-17*
Journal: *Nature Communications*
Ref. No.: *NCOMMS-23-00794-T*

- 2.3. “Moreover, in the methods section, it is essential to clarify how the children are treated once have received the first booster dose. Are they excluded from the analysis or the administration date is used as the index date?”**

Author Response:

Thank you for this helpful suggestion. We have now clarified that those receiving the third dose were censored upon their third dose vaccination, after which they would not be eligible to be sampled as cases nor controls.

“All observation periods were censored till the date of their third dose of vaccination”

(Lines 174-175, P. 8)

- 2.4. “In the matching procedure only two confounders are taken into account: sex and age. I would suggest matching also by the week of the administration of the first and second dose since this could be an important confounder in your analysis and could affect the final results.”**

Author Response:

Thank you for your suggestion. We have indeed considered matching cases and controls by the time duration since the date of receiving the second dose. However, this would inevitably render the sampling frame for the controls much narrower given our modest sample size of this study population. We have therefore decided to adjust for this important confounding factor by including it as one of the covariates in the multivariable conditional logistic regression. We believe it should have been properly addressed with this approach.

- 2.5. “An important point is the use of the hospitalization date as the index date. I would suggest using the infection date as the index date for the hospitalized cases. In the estimation of the protection against COVID-19 hospitalization, it is important to include all the infections resulting in hospital admission within 28 days in the study period. Using the hospitalization date, you could risk defining as hospitalized case individuals infected before the 1st January and on the other hand, you would risk excluding from the case definition individuals who developed the infection before the end of the study (15 August).”**

Author Response:

Response to Reviewer Comments

Manuscript Title: *Effectiveness of BNT162b2 after extending the primary-series dosing interval in children and adolescents aged 3-17*
Journal: *Nature Communications*
Ref. No.: *NCOMMS-23-00794-T*

Thank you for your suggestion. We have run an additional sensitivity analysis using the date of infection for the hospitalization outcome [AOR: 0.747 (0.34, 1.641)]. Results remain consistent with the main results.

2.6. “Please, clarify if individuals previously infected are excluded from the underlying cohort.”

Author Response:

Thank you for this helpful suggestion. We have now clarified that those receiving the third dose were censored upon their third dose vaccination, after which they would not be eligible to be sampled as cases nor controls.

“All observation periods were censored till the date of their third dose of vaccination.”

(Lines 174-175, P. 8)

2.7. “Please, detail the flowchart including the number of individuals that move from the control to the case group.”

Author Response:

Thank you very much. Please find the number of controls being subsequently included as cases now reported in the Results section.

“3.0% (653/21577) of controls in the Covid-19 matched set, and 0.1% (1/807) controls in Covid-19 related hospitalizations matched set subsequently developed into cases.”

(Lines 66-68, P. 4)

2.8. “I would suggest changing the caption of the tables because they do not fully describe the content of the tables”

Author Response:

Thank you very much. We have now edited the table captions for your further comments. The caption of the tables has changed to “*Table 1. Characteristics of cases and controls in two case-control studies for Covid-19 infection and Covid-19 hospitalization*” and “*Table 2. The risk of Covid-19 infection and hospitalization for*

Response to Reviewer Comments

Manuscript Title: *Effectiveness of BNT162b2 after extending the primary-series dosing interval in children and adolescents aged 3-17*

Journal: *Nature Communications*

Ref. No.: *NCOMMS-23-00794-T*

children and adolescents with extended dosing intervals compared with regular dosing intervals”.

2.9. “I found several typos and writing mistakes I would suggest asking a native English speaker to review the manuscript”

Author Response:

Thank you very much. We have sought professional help to proofread the manuscript and minimize errors. We believe the language and reporting styles of the manuscript are now up to publication standard. We have acknowledged the help received from the proof-reader under acknowledgements.

Reviewer #3

3.1. “This article addresses an important topical issue of what is the optimal dosing interval of COVID-19 vaccines for children and adolescent. The insights provided by this article provides guidance to policy makers in setting optimal dosing intervals”

Author Response:

Thank you very much for your positive comment on the importance of this work.

3.2. “There are 2 main weaknesses in the use of observational data which exploits a potential policy change. The first is an issue of selection. Those who chose to receive their vaccinations earlier where the 28 dose-interval was in force, are likely those who think that their children are at higher risk of contracting COVID. The results that the authors are picking up may entirely be due to the higher exposure risk of those who chose to be vaccinated earlier. The article should be clearer about whether when there was a chance of policy of moving from a 28 day to 56 dose recommended dosing interval. It would be useful to highlight what was the vaccination rates at various points of policy shifts. This would allow the reader to assess the degree of selection bias that might be driving the results.”

Author Response:

Thank you for this comment. However, we believe this issue has been appropriately addressed with our current approach. We conducted an incidence density sampling for the selection of cases and controls, whereby all the participants in the same

Response to Reviewer Comments

Manuscript Title: *Effectiveness of BNT162b2 after extending the primary-series dosing interval in children and adolescents aged 3-17*

Journal: *Nature Communications*

Ref. No.: *NCOMMS-23-00794-T*

matched set were exposed to the same degree of Covid-19 risks because they were observed on the exact same day, i.e., the index date. The comparison is, therefore, completely fair in this regard. The main independent variable of interest, i.e., different priming dosing intervals, could certainly occur at different time periods, but the fact that we have adjusted the time since the second dose in the conditional logistic regression analysis should already be sufficient to account for the waning of the protection effect from the priming doses over time. We have now highlighted in the Methods section how this approach could enable us to compare cases and controls exposed to the same level of Covid-19 risk.

“By using incidence density sampling, all participants in the same matched set were exposed to the same degree of Covid-19 risk and were under the same policy guidance.”

(Lines 181-183, P. 8)

- 3.3. “Second, given that those who had a 28-day dose interval received their vaccinations earlier, the authors may simply be picking up waning in vaccine effectiveness, rather than the better performance of an extended dosing intervals. Here too, authors ought to match by time from second dose (i.e. i.e. difference between the index date and the date of second dose). If this is not possible, then the time from second dose ought to be reported in summary statistics for the cases and controls, for different dose intervals.”**

Author Response:

Thank you for this comment. As mentioned in our response to Comment 3.2, we have adjusted the time since the second dose in the conditional logistic regression analysis and it should already be sufficient to account for the waning of the protection effect from the priming doses over time. At the earlier stages of this study, we indeed considered matching cases and controls by the time duration since the date of receiving the second dose. However, this would inevitably render the sampling frame for the controls much narrower, given our modest sample size of this study population.

We hope these proposed revisions and additional analyses sufficiently address the concerns raised by the reviewers. We look forward to hearing from you again.

Sincerely,

Esther Wai Yin Chan, PhD

Corresponding author

Response to Reviewer Comments

Manuscript Title: *Effectiveness of BNT162b2 after extending the primary-series dosing interval in children and adolescents aged 3-17*
Journal: *Nature Communications*
Ref. No.: *NCOMMS-23-00794-T*

Associate Professor
Department of Pharmacology and Pharmacy
Li Ka Shing Faculty of Medicine
The University of Hong Kong

Response to Reviewer Comments

Manuscript Title: *Effectiveness of BNT162b2 after extending the primary-series dosing interval in children and adolescents aged 3-17*

Journal: *Nature Communications*

Ref. No.: *NCOMMS-23-00794-T*

Reference

1. HKSAR Government. *SARS-CoV-2 variants*. 2022 19 July 2022; Available from: <https://www.chp.gov.hk/en/statistics/data/10/641/100135/6973.html>.
2. HKSAR Government. *COVID-19 Vaccination Programme opens to persons aged 16 or above*. 2021 April 15, 2021 [cited 2022 November 28]; Available from: <https://www.info.gov.hk/gia/general/202104/15/P2021041500565.htm>.
3. HKSAR Government. *Persons aged 12 to 15 can make reservations to receive BioNTech vaccine from tomorrow*. 2021 June 10, 2021 [cited 2022 October 1,]; Available from: <https://www.info.gov.hk/gia/general/202106/10/P2021061000556.htm>.
4. HKSAR Government. *Arrangements for children aged 5 to 11 to receive COVID-19 vaccines*. 2022 January 20, 2022 [cited 2022 November 28]; Available from: <https://www.info.gov.hk/gia/general/202201/20/P2022012000714.htm>.
5. HKSAR Government. *COVID-19 vaccination of children aged under 3 begins today*. 2022 August 4, 2022 [cited 2022 November 28]; Available from: <https://www.info.gov.hk/gia/general/202208/04/P2022080400470.htm>.

REVIEWER COMMENTS

Reviewer #1 (Remarks to the Author):

Thank you for your thoughtful and comprehensive response to initial peer review. I am satisfied by the changes, but would strongly suggest that you remove reference to the <5 year olds throughout, as they were not eligible for two doses during the study period.

Reviewer #2 (Remarks to the Author):

I would like to thank the authors for the effort in improving the quality of the paper, however in my opinion there are some points that still need to be addressed before recommending the paper ready for publication:

1. Please clarify in the introduction the main steps of the vaccination campaign in the study population: time, dosages of BNT162b2, number of recommended doses
2. Analysing Table 2 in the Appendix (number of cases and controls by age group), I would suggest clarifying in the title and the abstract that the study is focused on children and adolescents 5-17. Although 3-4 year-olds were eligible for vaccination, there are no data on this age group. Moreover, as highlighted in the discussion, children aged 3-11 and 12-17 are administered different vaccination dosages. To address the necessity of specific policy recommendations, I suggest doing a stratified analysis at least for the group 12-17 years since the sample size should not be a problem for this age group. Several studies have already shown that the vaccine effectiveness for these age groups is strongly different ([https://www.thelancet.com/journals/lancet/article/PIIS0140-6736\(22\)01185-0/fulltext](https://www.thelancet.com/journals/lancet/article/PIIS0140-6736(22)01185-0/fulltext;); <https://www.nejm.org/doi/full/10.1056/NEJMoa2210058>). This point should be discussed more in-depth.

Reviewer #3 (Remarks to the Author):

Thank you for addressing my comments in 3.3 of your response, and I am satisfied with the response here.

However, I don't think my comments in 3.2 were adequately addressed. I do understand that you are analysing based on the same dates of exposure and are controlling for force of infection. What I am raising is the issue of self-selection, that families that vaccinated their kids earlier when the 21 day regime is in force, and at higher risk groups (i.e. going to larger schools, parents are working etc) as compared to those who were vaccinated later when the 56 days regime is in force. This self-selection is resulting in a spurious positive result that you have identified.

This is NOT easy to correct for, so it may be that authors need to accept this as a limitation in the discussion.

However, I think the following would be useful.

1. In this sentence, to highlight when exactly the change occurred, and what is the proportion of children vaccinated by this time.
"Similar to other jurisdictions, Hong Kong took early action in extending the recommended dosing interval between the first two doses(7), i.e. primary series, of BNT162b2 from 21 days to 56 days for adolescents under the age of 18 to reduce the risk of myocarditis (8-11).
2. Comparison using summary statistics of those who were vaccinated early versus later just to make sure that they are not systematic differences in the comparison group.
3. Including as a sensitivity analysis, a subsegment analysis on those who received their second

dose say 21 or 30 days before/after the policy cutoff. We are trying to restrict here to people who wanted their children to get vaccinated at about the same time. CIs for this are likely to be large, but I think if point estimates are in the ballpark of the main analysis, would allow us to at least say that self-selection may not be so significant.

Thank you very much for further considering our manuscript. We are very thankful for the highly constructive reviews relayed from the referees. Please find appended below our detailed response to each of their specific remaining comments with proposed changes. The revised relevant text is quoted as appropriate:

Reviewer #1

- 1.1. **“Thank you for your thoughtful and comprehensive response to initial peer review. I am satisfied by the changes, but would strongly suggest that you remove reference to the <5 year olds throughout, as they were not eligible for two doses during the study period.”**

Author Response:

Thank you very much for your encouraging overall comment on the revised manuscript. We have removed all references to children aged below five years throughout the text and changed the title of the manuscript as well.

“Effectiveness of BNT162b2 after extending the primary-series dosing interval in children and adolescents aged 5-17”

(Title)

“To examine this potential variable effectiveness, we conducted a population-based nested case-control study of children and adolescents aged 5-17 years who had received two doses of BNT162b2 in Hong Kong.”

(Lines 30-32, P. 2)

“Fifth, pediatric formulations were used specifically for children aged younger than 12 years, which indicates a dosage difference in children aged 5-11 and 12-17.”

(Lines 136-137, P. 6)

Reviewer #2

Response to Reviewer Comments

Manuscript Title: *Effectiveness of BNT162b2 after extending the primary-series dosing interval in children and adolescents aged 3-17*
Journal: *Nature Communications*
Ref. No.: *NCOMMS-23-00794-A*

- 2.1. “I would like to thank the authors for the effort in improving the quality of the paper, however in my opinion there are some points that still need to be addressed before recommending the paper ready for publication”**

Author Response:

Thank you very much for the positive overall comment on our revised manuscript. We hope the following response suffice to address your remaining concerns.

- 2.2. “1. Please clarify in the introduction the main steps of the vaccination campaign in the study population: time, dosages of BNT162b2, number of recommended doses”**

Author Response:

Thank you for this comment. We have now detailed the key steps of the mass vaccination campaign for children and adolescents in the Introduction section.

“The BNT162b2 vaccination program for adolescents in Hong Kong was launched in June 2021. The age threshold was extended to ≥ 16 years on April 15, 2021, ≥ 12 on June 11, 2021, ≥ 5 on January 21, 2022 and subsequently, ≥ 6 months old on August 4, 2022. (7-10) ”

(Lines 46-48, P. 3)

- 2.3. “2. Analysing Table 2 in the Appendix (number of cases and controls by age group), I would suggest clarifying in the title and the abstract that the study is focused on children and adolescents 5-17. Although 3-4 year-olds were eligible for vaccination, there are no data on this age group.”**

Author Response:

Thank you for this helpful suggestion. We have now changed the title and abstract together with all previous references to children aged three or four to clarify this point.

“Effectiveness of BNT162b2 after extending the primary-series dosing interval in children and adolescents aged 5-17”

(Title)

Response to Reviewer Comments

Manuscript Title: *Effectiveness of BNT162b2 after extending the primary-series dosing interval in children and adolescents aged 3-17*
Journal: *Nature Communications*
Ref. No.: *NCOMMS-23-00794-A*

“To examine this potential variable effectiveness, we conducted a population-based nested case-control study of children and adolescents aged 5-17 years who had received two doses of BNT162b2 in Hong Kong.”

(Lines 30-32, P. 2)

“Fifth, pediatric formulations were used specifically for children aged younger than 12 years, which represent a dosage difference between children aged 5-11 and 12-17.”

(Lines 136-137, P. 6)

- 2.4. “Moreover, as highlighted in the discussion, children aged 3-11 and 12-17 are administered different vaccination dosages. To address the necessity of specific policy recommendations, I suggest doing a stratified analysis at least for the group 12-17 years since the sample size should not be a problem for this age group. Several studies have already shown that the vaccine effectiveness for these age groups is strongly different ([https://www.thelancet.com/journals/lancet/article/PIIS0140-6736\(22\)01185-0/fulltext](https://www.thelancet.com/journals/lancet/article/PIIS0140-6736(22)01185-0/fulltext); <https://www.nejm.org/doi/full/10.1056/NEJMoa2210058>). This point should be discussed more in-depth.”**

Author Response:

Thank you for your suggestion. We have now conducted a stratified analyses including only those aged 12-17 and the findings are largely consistent with the main results.

“Subgroup analysis by sex and among those aged 12-17 were conducted”

(Lines 196-197, P. 9)

“A subgroup analysis for adolescents between 12-17 years old estimated a similar odds ratio with the primary analysis, for both Covid-19 risk (aOR 0.724, 95%CI: 0.624, 0.840) and Covid-19-related hospitalization (aOR 0.835, 95%CI: 0.360,1.939).”

(Lines 79-81, P. 4)

Reviewer #3

- 3.1. “Thank you for addressing my comments in 3.3 of your response, and I am satisfied with the response here.”**

Response to Reviewer Comments

Manuscript Title: *Effectiveness of BNT162b2 after extending the primary-series dosing interval in children and adolescents aged 3-17*

Journal: *Nature Communications*

Ref. No.: *NCOMMS-23-00794-A*

Author Response:

Thank you very much for your positive comment on our response to Comment 3.3.

- 3.2. “However, I don't think my comments in 3.2 were adequately addressed. I do understand that you are analysing based on the same dates of exposure and are controlling for force of infection. What I am raising is the issue of self-selection, that families that vaccinated their kids earlier when the 21 day regime is in force, and at higher risk groups (i.e. going to larger schools, parents are working etc) as compared to those who were vaccinated later when the 56 days regime is in force. This self-selection is resulting in a spurious positive result that you have identified. This is NOT easy to correct for, so it may be that authors need to accept this as a limitation in the discussion.”**

Author Response:

Thank you for this comment. We agree there may be a potential confounding effect from a self-selection process arising from the potential differences between those who receive a regular dosing interval and those who receive an extended dosing interval. We have accordingly included this as a limitation.

“Fifth, there may be potential differences in various unmeasured characteristics between families who chose the regular dosing interval and those who chose the extended dosing interval by self-selection.”

(Lines 141-143, P. 7)

- 3.3. “1. In this sentence, to highlight when exactly the change occurred, and what is the proportion of children vaccinated by this time. "Similar to other jurisdictions, Hong Kong took early action in extending the recommended dosing interval between the first two doses(7), i.e. primary series, of BNT162b2 from 21 days to 56 days for adolescents under the age of 18 to reduce the risk of myocarditis (8-11).”**

Author Response:

Thank you for this comment. We have now specified the exact date this policy change occurred.

Response to Reviewer Comments

Manuscript Title: *Effectiveness of BNT162b2 after extending the primary-series dosing interval in children and adolescents aged 3-17*
Journal: *Nature Communications*
Ref. No.: *NCOMMS-23-00794-A*

“Similar to other jurisdictions, Hong Kong took early action in extending the recommended dosing interval between the first two doses(11), i.e. primary series, of BNT162b2 from 21 days to 56 days for adolescents under the age of 18 to reduce the risk of myocarditis on June 17, 2022”

(Lines 48-51, P. 3)

- 3.4. “2. Comparison using summary statistics of those who were vaccinated early versus later just to make sure that they are not systematic differences in the comparison group.”**

Author Response:

Thank you for this comment. We have now created a supplementary table in the appendix to show the characteristics of the participants stratified by case-control status as well as exposure status. Overall, we did not find notable differences across exposure status except the onset time from second dose to events of interests, which has been properly adjusted in the model.

“Detailed demographic information stratified by both case and exposure status is shown in Appendix 2.”

(Lines 70-71, P. 4)

- 3.5. “3. Including as a sensitivity analysis, a subsegment analysis on those who received their second dose say 21 or 30 days before/after the policy cutoff. We are trying to restrict here to people who wanted their children to get vaccinated at about the same time. CIs for this are likely to be large, but I think if point estimates are in the ballpark of the main analysis, would allow us to at least say that self-selection may not be so significant.”**

Author Response:

Thank you for this suggestion. We have now conducted a new sensitivity analysis whereby the exposure status was categorized into 1. regular dosing interval with the second dose being received 21 days or more before the policy change; 2. regular dosing interval with the second dose being received within <21 days before the policy change or after it; 3. extended dosing interval with the second dose being received 21 days or more before the policy change; 4. extended dosing interval with the second dose being received within <21 days before, or after the policy change. We did not identify notable differences in the results compared with the main analysis.

Response to Reviewer Comments

Manuscript Title: *Effectiveness of BNT162b2 after extending the primary-series dosing interval in children and adolescents aged 3-17*
Journal: *Nature Communications*
Ref. No.: *NCOMMS-23-00794-A*

“...5) The exposure statuses were further categorized by dosing interval (extended versus regular) and whether the second dose was received 21 days or more before the policy change to identify any potential confounding effects from the policy change, independent of the effect from the modified regimen”
(Lines 201-204, P. 10)

“A similar protective effect was evident when examining only patients with extended dosing intervals who received their second dose 21 days or more before the policy change (aOR: 0.690, 95%CI: 0.593, 0.802 for covid-19 infection and aOR: 0.716, 95%CI:0.313, 1.638 for covid-19-related hospitalization).”

(Lines 89-92, P. 5)

We thank the reviewers for their valuable perspectives and we hope these proposed revisions and additional analyses sufficiently addresses the remaining concerns We look forward to hearing from you again.

Sincerely,

Esther Wai Yin Chan, PhD
Corresponding author

Associate Professor
Department of Pharmacology and Pharmacy
Li Ka Shing Faculty of Medicine
The University of Hong Kong